# Parallel Losses of Blue Opsin Correlate with Compensatory Neofunctionalization of UV-Opsin Gene Duplicates in Aphids and Planthoppers

**DOI:** 10.3390/insects14090774

**Published:** 2023-09-20

**Authors:** Markus Friedrich

**Affiliations:** 1Department of Biological Sciences, Wayne State University, 5047 Gullen Mall, Detroit, MI 48202, USA; friedrichm@wayne.edu; 2Department of Ophthalmological, Visual, and Anatomical Sciences, School of Medicine, Wayne State University, 540 East Canfield Avenue, Detroit, MI 48201, USA

**Keywords:** Hemiptera, opsin, gene loss, color vision, compensatory neofunctionalization, tuning site

## Abstract

**Simple Summary:**

Hemiptera is one of the largest and most diverse insect orders, yet our knowledge about visual opsin evolution in this group remains preliminary. This study provides an updated survey of visual opsin genes in the Hemiptera, which reveals that the subfamily of blue-sensitive opsin receptor genes has been independently lost in planthoppers and aphids. Moreover, in both groups, tandem duplication of UV-sensitive opsins produced sister paralogs that diverged by maintaining ancestral UV sensitivity versus shifting to blue sensitivity and thereby likely compensating for the loss of the blue-sensitive opsin subfamily. The study further shows that these parallel trajectories at the level of gene family evolution are associated with mostly convergent changes at the level of protein sequence evolution.

**Abstract:**

Expanding on previous efforts to survey the visual opsin repertoires of the Hemiptera, this study confirms that homologs of the UV- and LW-opsin subfamilies are conserved in all Hemiptera, while the B-opsin subfamily is missing from the Heteroptera and subgroups of the Sternorrhyncha and Auchenorrhyncha, i.e., aphids (Aphidoidea) and planthoppers (Fulgoroidea), respectively. Unlike in the Heteroptera, which are characterized by multiple independent expansions of the LW-opsin subfamily, the lack of B-opsin correlates with the presence of tandem-duplicated UV-opsins in aphids and planthoppers. Available data on organismal wavelength sensitivities and retinal gene expression patterns lead to the conclusion that, in both groups, one UV-opsin paralog shifted from ancestral UV peak sensitivity to derived blue sensitivity, likely compensating for the lost B-opsin. Two parallel bona fide tuning site substitutions compare to 18 non-corresponding amino acid replacements in the blue-shifted UV-opsin paralogs of aphids and planthoppers. Most notably, while the aphid blue-shifted UV-opsin clade is characterized by a replacement substitution at one of the best-documented UV/blue tuning sites (Rhodopsin site 90), the planthopper blue-shifted UV-opsin paralogs retained the ancestral lysine at this position. Combined, the new findings identify aphid and planthopper UV-opsins as a new valuable data sample for studying adaptive opsin evolution.

## 1. Introduction

Public awareness of hemipteran insects is largely due to the detrimental impact of pest species, such as aphids and stink bugs, or the horrifying experience of exposure to bed bugs and alike [1,2,3]. Hemiptera, i.e., true bugs, however, diversified into far more than human nuisances. Its members conquered terrestrial and aquatic biotas by extracting nutrients from a diverse panel of animal and plant species with the help of sucking mouthparts, the morphological hallmark of this clade [4,5,6]. Select groups evolved traits as remarkable as acoustic long-distance communication (cicadas), water sliding (water striders), and diving (backswimmers and many other families). Our familiarity with hemipteran insects is also due to their expansive number of species. With over 95,000 documented species, the Hemiptera are the largest order of direct-developing insects. In fact, their abundance is only challenged by the four megadiverse orders of holometabolous insects, i.e., the Coleoptera (beetles, ~350,000 species), Diptera (true flies, ~160,000 species), Hymenoptera (bees and wasps, ~150,000 species), and Lepidoptera (moths and butterflies, ~150,000 species) [7].

Phylogenomic studies have begun to consolidate our understanding of hemipteran diversification [8,9,10], confirming the existence of three large suborders: the Sternorrhyncha (~12,500 species of aphids, whiteflies, and scale insects), Auchenorrhyncha (~42,000 species of cicadas, leafhoppers, treehoppers, planthoppers, and spittlebugs), and the Heteroptera (~45,000 species). The latter include families of well-known terrestrial species, like the kissing bugs (Reduviidae), bedbugs (Cimicidae), plant bugs (Miridae), seed bugs (Lygaeidae), and stink bugs (Pentatomidae and related families), but also aquatic taxa, such as water striders (Gerridae) and backswimmers (Notonectidae).

First insights into the genomic dimensions of hemipteran diversification have been gained through the genome sequence assemblies for a number of key species [4], starting with the pea aphid *Acyrtosiphon pisum* [11], the kissing bug *Rhodnius prolixus* [3], and the bed bug *Cimex lectularius* [1]. Subsequent efforts added genomes from specialist species, like the water strider *Gerris buenoi* [12], the laboratory model *Oncopeltus fasciatus* [13], and the closely related pentatomid stink bug *Halyomorpha halys* [14]. Today, over 80 hemipteran genome assemblies are available through the National Center for Biotechnology Information (NCBI) [15] for deeper studies of gene families of interest.

The members of the opsin gene family encode light-sensitive G-protein-coupled transmembrane receptors (GPCRs). Their capacity to absorb photon energy functions at the beginning of the phototransduction cascade in animal photoreceptors [16]. Paralogs of this ancient sensory gene family differ by wavelength-dictated photon absorption maxima, which have been optimized by natural selection to provide detection sensitivity to fitness-affecting spectra of the environment. In many cases, this has also led to the capacity to differentiate between wavelength inputs, i.e., color vision [17]. Insects are characterized by three opsin subfamilies whose members are expressed in the photoreceptors of the peripheral visual system, i.e., visual opsins expressed in the compound eyes and the ocelli [18,19,20,21]. This includes the subfamily of green or long wavelength (LW)-sensitive opsins with sensitivity peaks centered around 520 nm, the blue (B) sensitive opsins with sensitivity maxima around 450 nm, and UV-sensitive opsins with sensitivity maxima between 360 nm and 380 nm [18].

Color vision frequently correlates with the conservation of all three of these canonical opsin subfamilies. Crepuscular or nocturnal species, however, are often characterized by the possession of UV- and LW-opsins only, i.e., the evolutionary loss of the B-opsin subfamily. Hemipteran examples of this paradigm include the kissing bug and the bed bug [1]. Less expected, however, was the genomic evidence of the lack of B-opsins in the pea aphid, *A. pisum* [11,22]. Equally surprising, the genome-wide analyses of the opsin repertoires of the water strider, *G. buenoi*, and the milkweed bug, *O. fasciatus*, also failed to detect B-opsins [12,13]. Consistent with a general absence of B-opsins in the Heteroptera [23] reported the conservation of LW- and UV-opsins but an absence of B-opsin in plant bugs (Heteroptera: Miridae). Moreover, their finding of two LW-opsin paralogs in the plant bugs based on transcriptome evidence aligned well with the earlier reported expansion of the LW-opsin subfamily in the Miridae based on a Heteroptera-wide compilation of genomic and transcriptomic opsin sequences [14].

While the lack of B-opsins from hemipteran taxa as distantly related as aphids and Heteroptera raised the possibility of an early loss of this subfamily during hemipteran diversification, the documentation of B-opsin in the green rice leafhopper *Nephotettix cincticeps* (Auchenorrhyncha: Cicadomorpha) pointed to a more complex scenario [24]. Moreover, transcriptome searches recovered a B-opsin homolog in the hackberry petiole gall psyllid *Pachypsylla venusta* (Sternorrhyncha) [12]. Corroborating the conservation of B-opsin in the Sternorrhyncha, a B-opsin homolog was recently also reported from the Asian citrus psyllid *Diaphorina citri* [25]. Combined, these data indicate that the B-opsin subfamily was conserved in early Hemiptera and lost multiple times during later diversification in select subgroups.

In contrast to this scenario in the Hemiptera, the vast species diversity of the insect order Coleoptera has been discovered to consistently lack B-opsins due to the likely loss of this gene family in the lineage of the Neuropteroidea [26,27]. A compelling case, however, has been made that many coleopteran lineages likely regained trichromacy through gene duplications in the LW- and UV-opsin subfamilies [26]. Intriguingly, UV-opsin tandem gene duplicates have also been discovered in aphids [22]. Moreover, physiological studies unearthed evidence of blue sensitivity in aphids [28]. Similarly, the greenhouse whitefly *Trialeurodes vaporariorum*, a member of the Sternorrhyncha, was identified as blue-sensitive [29] Taken together, these data suggest that aphids and planthoppers might have independently retained or regained blue sensitivity by compensatory gene duplication in the process of, or after, losing the B-opsin subfamily.

Here, I present the results from a more extensive analysis of B-opsin conservation in the Hemiptera. This effort corroborates the existence of B-opsins in select lineages of the Sternorrhyncha and Auchenorrhyncha. In addition, B-opsin was also recovered from a representative of the moss bugs, i.e., the smaller hemipteran suborder Coleorrhyncha. The survey further reveals that the lack of B-opsin in subclades of the Sternorrhyncha and Auchenorrhyncha, i.e., in aphids (Aphidoidea) and planthoppers (Fulgoroidea), respectively, is correlated with the presence of tandem-duplicated UV-opsins. Protein sequence evolution evidence, currently available wavelength sensitivity data, and data on retinal expression patterns converge to suggest that, in both planthoppers and aphids, one clade of UV-opsin sister paralogs neofunctionalized by shifting from ancestral UV- to derived maximum blue sensitivity while the other clade retained ancestral UV-sensitivity. These remarkably similar trajectories contrast with the Heteroptera, where the lack of B-opsin is a trait of the entire taxon and is accompanied by several independent expansions of the LW-opsin subfamily. Preliminary probing for tuning substitutions in the neofunctionalized B-opsins of aphids and planthoppers suggests a higher number of lineage-specific than parallel amino acid replacement substitutions. Taken together, these findings highlight Hemiptera as a pivotal group for studying the adaptive evolution of opsin GPCRs in addition to the ongoing studies in Coleoptera, Lepidoptera, and Diptera [16,26,30,31,32,33,34].

## 2. Materials and Methods

### 2.1. Homolog Compilation

Previously identified visual opsin genes were extracted from the genome assembly reports of the stink bug *H. halys* [14], the mikweed bug *O. fasciatus* [13], the water strider *G. buenoi* [12], the bed louse *C. lectularius* [1], the kissing bug *R. prolixus* [2], and the pea aphid *A. pisum* [11,35]. Additional genome assemblies interrogated for hemipteran visual opsins included that of the brown planthopper species *Nilaparvata lugens* [36,37] and *Laodelphax striatellus* [38], the green leafhopper *N. cincticeps* [39], the Cedar bark aphid *Cinara cedri* [40], the Asian citrus psyllid *D. citri* [41], and the silverleaf whitefly *Bemisia tabaci* [42].

Further homologs were identified through BLASTp searches of the NCBI non-redundant (nr) protein sequence database and tBLASTn searches of the NCBI Transcriptome Shotgun Assembly (TSA) database [15,43,44].

### 2.2. Sequence Analyses

Protein sequence translations of identified opsin transcripts were produced with “Translate tool” of the Expasy suite hosted by the Swiss Institute of Bioinformatics [45]. Evolutionary protein sequence divergences were estimated with the MEGA software package [46] using the Jones–Taylor–Thornton (JTT) model and applying a gamma parameter value of 1. Pairwise relative rate tests [47] and maximum likelihood tests of molecular clock constancy were likewise conducted in the MEGA suite [46] on multiple sequence alignments generated with T-Coffee [48] followed by removal of ambiguous sites with TrimAL [49] using the “gappyout” stringency options.

### 2.3. Gene Tree Reconstruction

The global opsin protein sequence gene tree was estimated with RAxML [50] hosted in the CIPRES Science Gateway environment [51] from a multiple sequence alignment produced by T-Coffee [48] and cleared of ambiguous regions applying Gblocks default settings [52].

## 3. Results

### 3.1. Singleton B-Opsin Homologs in Sternorrhyncha, Auchenorrhyncha, and Coleorrhyncha

To explore the pervasiveness of B-opsin conservation in the Sternorrhyncha and Auchenorrhyncha, and to further scrutinize their absence from the Heteroptera, I used the B-opsin homologs of *N. cincticeps* [24] and *D. citri* [25] as queries in BLASTp and tBLASTn searches of current gene, transcript, and genome assembly databases (Appendix A). B-opsin family membership was probed by BLAST searches of candidate homologs against the *Drosophila melanogaster* genome assembly (NCBI Release 6.46) and analysis of relationships in a maximum likelihood estimated gene tree (Figure 1). These efforts identified singleton B-opsin homologs in additional Sternorrhyncha and Auchenorrhyncha species. In the Sternorrhyncha, B-opsins were detected in the hackberry nipplegall maker *Pachypsylla celtidismamma* and the silverleaf whitefly *B. tabaci* (Appendix A and Figure 1). In the Auchenorrhyncha, BLAST searches uncovered B-opsins in the green leafhopper *Empoasca vitis*, the tea green leafhopper *Matsumurasca onukii*, the spittlebug *Clastoptera arizonana*, and the cicada species *Tamasa doddi* and *Burbunga queenslandica* (Appendix A and Figure 1). Moreover, B-opsin was also recovered from *Xenophysella greensladeae*, a member of the relatively small but likewise ancient hemipteran suborder Coleorrhyncha (moss bugs) (Figure 1).

Replicating the outcome of previous efforts, no B-opsins were recovered from heteropteran species [12,14]. Taken together, these findings confirmed the ancestral conservation of the B-opsin gene family in the Hemiptera and documented its continued conservation in the Coleorrhyncha besides select lineages of the Sternorrhyncha and Auchenorrhyncha.

### 3.2. Parallel Losses of B-Opsin in the Heteroptera, Auchenorrhyncha, and Sternorrhyncha

Parsimony-guided manual gene-species tree reconciliation [53] mapped the loss of B-opsin in the Heteroptera to the stem lineage of this group, as previously suggested [14] (Figure 2). Similarly consistent with previous conclusions, the global opsin gene tree also reproduced the parallel expansions of the LW-opsin gene family in the Gerridae and Pentatomidae (Figure 2) [14].

In the Auchenorrhyncha, the exclusive conservation of B-opsins in members of the Cicadomorpha led to the conclusion that B-opsin was selectively lost in the lineage to the Fulgoroidea, i.e., planthoppers, somewhere between 200 and 300 million years ago (Figure 2). In the Sternorrhyncha, the conservation of B-opsin in the Psylloidea and Aleyrodidae suggested that the consistent lack of B-opsin in the five sampled aphid species was due to its loss in the lineage to Aphidoidea, at least ~100 million years ago (Figure 2). Overall, these findings located a minimum of three parallel B-opsin losses in the hemipteran tree of life.

### 3.3. B-Opsin Losses Correlate with UV-Opsin Duplications in the Aphids and Planthoppers

The second insight emerging from the opsin compilation survey and gene-species tree reconciliation was the correlation between B-opsin conservation and the number of UV- or LW-opsin homologs (Table 1 and Figure 2). As noted above, many heteropteran clades are characterized by the possession of tandem duplicated LW-opsin genes (Figure 2) [12,14]. Contrasting this pattern, Auchenorrhyncha and Sternorrhyncha species that lack B-opsins, i.e., planthoppers and aphids, were characterized by duplicated UV-opsin genes (Table 1 and Figure 2).

Previous studies reported the tandem linkage of the duplicated UV-opsins in the pea aphid *A. pisum* [22]. The same constellation was encountered in other aphid species for which genome assemblies were available (Table 1). Even more notable, it was also the case in the genome assembly of the B-opsin-lacking member of the Auchenorrhyncha, the brown planthopper *N. lugens* (Table 1). In the aphid species, interlocus distances ranged from 21,000 bp to 30,000 bp (Table 1). In *N. lugens*, the UV-opsin sister paralogs were separated by less than 6000 bp.

### 3.4. Accelerated Protein Sequence Evolution Rates in the Putatively Blue-Shifted UV-Opsin Paralogs of Aphids and Planthoppers

A strong correlation between B-opsin losses and expansions in the LW- or UV-opsin gene families has also been found in beetles [26]. In this case, protein sequence analyses and experimental studies support the model that blue sensitivity is restored by the duplication and compensatory modification of LW- or UV-opsin homologs [54]. Intriguingly, the green peach aphid *M. persicae* was found to be green, blue, and UV-sensitive [28] despite its lack of B-opsin (Table 1). As a first step to explore the possibly compensatory nature of the UV-opsin duplications in the B-opsin-lacking aphids and planthoppers, I probed for unequal protein sequence diversification rates. As one UV-opsin could be hypothesized to maintain ancestral UV peak sensitivity while the other paralog shifted to blue sensitivity as a derived trait, ancestrally functioning UV-opsin paralogs were expected to differ by fewer amino acid differences from the conserved singleton UV-opsins of B-opsin possessing outgroup species.

In support of this scenario, for both the B-opsin lacking aphid and planthopper species, the duplicated UV-opsin homologs clustered into two paralog clades in the global and suborder-specific opsin gene trees (Figure 1 and Figure 3). In each case, one UV-opsin paralog clade was characterized by a longer basal branch than the sister paralog cluster (Figure 3). Consistent with these gene tree topologies, analysis of pairwise protein sequence differences corroborated the distinction between presumptive ancestral (UV-anc) and derived (UV-der) sister paralog clusters. In the planthoppers, the UV-anc opsin paralogs differed, on average, by 0.502 (±0.005) substitutions per site from the singleton UV-opsin of *M. onukii* compared to 0.55 (±−0.006) per site for the UV-der opsin paralogs. In the aphid species, the UV-anc opsin paralogs differed on average by 0.67 (±−0.03) substitutions per site from the singleton UV-opsin of *D. citri* compared to 0.75 (±−0.01) for the UV-der opsin paralogs (Appendix A).

Relative rate tests, finally, rejected substitution rate constancy when UV-anc opsin and UV-der opsin homologs were tested combined for the aphids (Figure 3a) or planthoppers (Figure 3b). When tested separately, however, rate constancy was not rejected for the orthologous members in the planthopper UV-anc and UV-der paralog clusters (Figure 3b). The same was true for the aphid UV-anc paralog cluster (Figure 3a). However, rate constancy was marginally rejected for the aphid UV-anc orthologs (Figure 3a), indicating more diversified substitution rates compared to the orthologs in the aphid UV-anc, planthopper UV-anc, and UV-der homolog groups.

### 3.5. Protein Sequence Change at Documented Tuning Sites

At the molecular level, the wavelength sensitivity peaks of specific opsin GPCRs are primarily determined by amino acid residues in the chromophore binding region, also called tuning sites [32]. Consistent with this mechanism, many studies have shown that amino acid residue changes at tuning sites diversified opsin wavelength specificities [54]. To probe the aphid and planthopper UV-der paralogs for tuning site substitutions, I surveyed 24 protein sequence sites that have been previously proposed or shown to represent tuning sites in UV-opsins of other insect orders [27,54].

In the aphid UV-opsin paralogs, consistent non-conservative amino acid residue differences between the UV-anc and UV-der opsin paralogs matched only two previously identified or proposed tuning sites, i.e., sites 90 and 105 as defined by homology to sites in the protein sequence of bovine Rhodopsin (Table 2 and Appendix A). Both of these sites were characterized by derived amino acid residue states in the UV-der opsin paralogs (Table 2). Similarly, in the planthopper UV-opsins, only two non-conservative amino acid residue differences between the UV-anc and UV-der opsin paralogs matched previously reported tuning sites. In this case, however, both the UV-anc paralog and the UV-der opsin paralog were each characterized by one derived amino acid residue state, i.e., at positions 293 and 294, respectively (Table 2).

While the overall number of amino acid residue changes in the previously predicted or demonstrated tuning sites was notably low, the replacement of positively charged lysine (K) with charge-neutral valine (V) at the site homologous to Rhodopsin amino acid residue 90 in the UV-der opsins of aphids was of exceptional interest, as previously noted [22]. Comparative data from both vertebrate and invertebrate species, as well as wavelength sensitivity studies of recombinant opsins in *Drosophila,* have shown that positively charged lysine at this position of the second GPCR transmembrane domain facilitates UV sensitivity while negatively charged or neutral amino acid residues such as asparagine (N) or glutamate (E) promote blue sensitivity [55]. Consistent with this paradigm, the singleton outgroup UV-opsins, and also the UV-anc opsin paralogs of both aphids and planthoppers, were occupied by lysine (K) at Rhodopsin site 90 (Table 2). The lysine (K) vs. valine (V) replacement at Rhodopsin site 90 in the UV-der opsins of aphids thus represented the highest confidence UV-to-blue sensitivity shift-related amino acid change in the surveyed UV-opsin homolog dataset. At the same time, the contrasting conservation of the ancestral UV-sensitivity-associated lysine state at Rhodopsin site 90 in both the UV-anc and UV-der opsin paralogs of planthoppers represented evidence of convergent trajectories toward blue sensitivity in the planthopper and aphid UV-opsins at the protein sequence level.

### 3.6. Parallel vs. Diversified Protein Sequence Changes

Given the evidence for B-opsin replacement through parallel UV-opsin duplicate neofunctionalization in the planthopper and aphids, scanning the UV-opsin paralogs of the two groups globally for parallel protein sequence changes offered the opportunity to identify new high-confidence tuning substitutions. In addition, this analysis was conducted to gauge the relative impacts of parallel vs. convergent protein sequence changes during the presumed adaptive transition from UV-to-blue sensitivity in the UV-der paralogs of aphids and planthoppers.

In the aphids, 11 sites were characterized by a derived non-conservative amino acid residue change in the UV-der paralogs vs. a single derived non-conservative amino acid residue change in the UV-anc paralogs (Table 2). Similarly, in the planthopper UV-opsins, 11 sites were characterized by derived non-conservative amino acid residue states in the UV-der paralogs. The number of derived non-conservative amino acid residue states in the UV-anc paralogs in the planthoppers, i.e., five, however, was notably higher compared with the aphid UV-anc paralogs (1) (Table 2). Most importantly, only two of the sites with derived non-conservative amino acid residue states matched between the UV-der paralogs of aphids and planthoppers (Table 2). 

One of them, site 47, mapped to the first GPCR transmembrane domain based on multiple sequence alignment correspondence [22]. Phenylalanine (F) was the ancestral amino acid residue state in both aphids and planthoppers. The UV-der opsins of aphids harbored a leucine (L) at this position, while the UV-der opsins of planthoppers carried a tyrosine (Y) (Table 2 and Appendix A). The other site was homologous to bovine Rhodopsin site 183, which resides in the second extracellular domain. Tyrosine (Y) represented the ancestral state in the UV-anc paralogs of both aphids and planthoppers, which was, in both groups, replaced by phenylalanine (F) in the UV-der paralogs (Table 2). Neither of the two sites, however, corresponded to previously predicted or demonstrated tuning sites (Table 2 and Appendix A).

Taken together, two protein sequence sites with bona fide parallel adaptive amino acid residue changes compared with 18 sites with non-shared putatively adaptive amino acid residue changes in the UV-der opsin paralogs of aphids and planthoppers. These findings provided further preliminary evidence that the parallel blue sensitivity shifts of the planthopper and aphid UV-der opsins were largely due to convergent evolution at the protein sequence level.

## 4. Discussion

Elucidating and explaining the diversification of opsin genes is at the center of current endeavors to gain deeper insights into the visual evolution of insects [16,18,26,31,32,33,34,56]. The present study adds two new examples of maintained trichromacy via the neofunctionalization of UV-opsin paralogs that compensate for the evolutionary loss of the B-opsin subfamily in distantly related clades of the Hemiptera.

### 4.1. At Least Three B-Opsin Losses during Hemipteran Diversification

Previous studies reported the lack of B-opsin throughout the Heteroptera [14], in the aphid species *A. pisum* and *M. persicae* [22] and in the planthopper *N. lugens* [24]. In isolation, these findings were compatible with a very early loss of the B-opsin gene family in the stem lineage of the Hemiptera. This straightforward scenario, however, was contradicted by the existence of B-opsin homologs in the green rice leafhopper *N. cincticeps* (Auchenorrhyncha: Cicadomorpha) [24], the hackberry petiole gall psyllid *P. venusta* (Sternorrhyncha) [12], and the Asian citrus psyllid *D. citri* [25]. The present study documents the existence of B-opsins in further taxa of the Sternorrhyncha (Aleyrodoidea, Psylloidea), Auchenorrhyncha (Cercopoidea, Cicadoidea, Membracoidea), and the Coleorrhyncha (moss bugs). Combined, these findings reveal that the B-opsin gene family had remained conserved in early Hemiptera but was subsequently lost at least three independent times.

Given the current limitation of taxon sampling density, it is possible that future studies will detect additional opsin gene family changes in younger hemipteran taxa. At this point, however, it seems reasonable to assume that the results presented here conclusively capture the number of B-opsin gene losses in the largest hemipteran suborders, i.e., the Heteroptera, Auchenorrhyncha, and Sternorrhyncha.

Some caution may be warranted concerning the actual number of B-opsin loss events in the Heteroptera. While the most parsimonious explanation for the absence of B-opsins in all Heteroptera is a single loss in the stem lineage of this group, less parsimonious scenarios of multiple early losses cannot be excluded with absolute certainty. In addition, there is evolutionary ancientness in these possible events, this is because of the considerable number of independent, and in some cases, very likely compensatory, LW-opsin duplications in this clade [14]. While resolving this ambiguity warrants further study, it is clear that the consequences of B-opsin loss in the Heteroptera differ in major interesting ways from aphids and planthoppers.

### 4.2. Timing B-Opsin Losses and UV-Opsin Duplications in Planthoppers and Aphids

In the aphids, the present study sampled five species from three subfamilies, i.e., the Aphidinae, Chaitophorinae, and Lachninae. Most molecular phylogenetic studies place these groups in distantly related branches of the aphid subfamily tree [57,58,59,60,61]. It, therefore, seems reasonable to postulate that the loss of B-opsin and correlated duplication of UV-opsin preceded the last common ancestor of aphids that has been estimated to have diversified about 100 million years ago [8]. At the same time, it remains possible that more comprehensive species sampling may produce evidence of a later loss of B-opsin in this group. Alternatively, the lack of B-opsins in aphids may even date back to a more ancient gene loss event, preceding the last common ancestor of aphids and scale insects (Coccoidea). Preliminary searches for B-opsin in this group failed to recover B-opsin homologs despite the availability of high-coverage genome assembly data [62]. However, sequence information from additional scale insect taxa will need to become available before firm conclusions can be drawn on this issue.

In the planthopper case, the conservation of B-opsin in the Coleorrhyncha and all other clades of Auchenorrhyncha besides planthoppers (Fulgoroidea), i.e., froghoppers (Cercopoidea), cicadas (Cicadoidea), and treehoppers (Membracoidea) makes it straightforward to map the loss of B-opsin to the early evolution of planthoppers [8]. However, only one of the 13 families of planthoppers was sampled in the present study, i.e., the Delphacidae. Thus, in this case, broader species sampling needs to be awaited before this B-opsin loss event can be dated with ultimate confidence.

Timing the parallel UV-opsin gene duplication events in aphids and planthoppers is equally essential to improve our understanding of whether their relationships to the B-opsin loss events are purely correlative or of a causative nature. Besides increasing the taxonomic depth of homolog sampling, molecular clock methods relating the divergencies of the UV-anc paralogs of planthoppers and aphids to calibrated UV-opsin split points in outgroup taxa might represent a complementary avenue to the more straightforward gene species tree reconciliation approach. At first glance, the tight physical linkages of the tandem duplicated UV-opsins might be interpreted to reflect evolutionary recentness (Table 2). This, however, is disputed by their conservation of at least 50 million years in both aphids and planthoppers. The deep conservation of close genetic linkages is, therefore, more likely reflecting constraints resulting from the shared use of cis-regulatory sequence elements. The parallel UV-opsin duplications in aphids and planthoppers constitute a valuable dataset to explore this possibility by investigating the existence and nature of conserved non-coding sequence space in the genomic vicinities of the tandem gene duplicates. 

### 4.3. Combined Evidence of B-Opsin Loss Compensation through Parallel UV-Opsin Neofunctionalization in Aphids and Planthoppers

Gene duplication events can lead to a variety of functionalization outcomes [63]. This includes the reduction of ancestral pleiotropy via differential inheritance of ancestral functions by the descendant paralogs, leading to subfunctionalization, or the acquisition of novel functions by one paralog, i.e., neofunctionalization. The latter outcome is frequently associated with dramatic protein sequence divergence between the emerging paralogs as one paralog changes more slowly under the continued constraints of ancestral functionality, while the second paralog experiences a burst of adaptive protein sequence change while optimizing to a new function. In the evolutionary diversification of opsin genes, the neofunctionalization scenario applies most clearly when a newly emerged paralog shifts its ancestral peak sensitivity. In support of this notion, peak sensitivity-shifted neofunctionalized opsin paralogs have been found to differ more dramatically from the singleton homologs in outgroup species than the paralog that preserves ancestral peak sensitivity [64]. Based on this criterion, one UV-opsin paralog, labeled UV-der, of each of the aphid and planthopper UV-opsin sister paralog pairs emerged as putatively wavelength sensitivity shifted, given its significantly higher overall amount of protein sequence change compared to the sister paralog. The latter, labeled UV-anc by the same logic, represents the candidate for maintaining ancestral UV sensitivity.

In the long term, it will be desirable to obtain definitive evidence of compensatory blue-shifted UV-opsin gene family paralogs in the planthoppers and aphids by in vitro spectral sensitivity analyses [54,65]. This will be particularly attractive for exploring the two new candidate tuning sites, 47 and 183, which are characterized by parallel amino acid substitutions in the aphid and planthopper UV-der paralogs. At this point, further support for the parallel compensatory UV-opsin neofunctionalization scenarios can be drawn from the findings of behavioral wavelength discrimination and electroretinogram (ERG) studies.

In the Sternorrhyncha, critical data are available from studies in the greenhouse whitefly *T. vaporariorum* [29], the pea aphid *A. pisum* [66], and the green peach aphid *M. persicae* [28]. Following up on earlier behavioral and ERG studies that produced evidence of UV- and LW-peak sensitivities in the greenhouse whitefly *T. vaporariorum* compound eyes [67], Stukenberg and Poehling (2019) demonstrated in high-resolution behavioral wavelength discrimination assays that *T. vaporariorum* is trichromatic with peak sensitivities in the UV (340–370 nm), blue (480–490 nm), and green (510–520 nm) [29]. Moreover, their data led to the model that blue sensitivity boosts the accuracy of settling onto green food plant targets. A key observation leading to this conclusion was the observation of an activity-inhibiting effect of exposure to blue light [29].

While sequence data are not available for *T. vaporariorum* at this time, its trichromacy aligns well with the ancestral possession of the B-opsin in the Aleyrodidae, as documented by the silverleaf whitefly *B. tabaci* (Table 1). The available ERG data for the B-opsin lacking *A. pisum* are considered ambiguous regarding the possibility of distinct spectral sensitivities in the blue and UV ranges [66]. ERG studies in the green peach aphid *M. persicae*, however, have shown that this species is characterized by a trichromatic profile of peak sensitivities, which match those of *T. vaporariorum,* despite the lack of B-opsin [28]. It thus seems safe to assume that the detected blue sensitivity in aphids is supplied by the UV-der opsin paralogs, as previously proposed [22].

In the Auchenorhyncha, behavioral and ERG data have been generated for the B-opsin-possessing green rice leafhopper *N. cincticeps* [68] and the B-opsin-lacking brown planthopper *N. lugens* [24]. Phototaxis assays revealed the highest attraction of *N. cincticeps* to green, blue, and yellow wavelengths [68]. ERG data revealed pronounced sensitivity peaks in the UV and LW ranges, but the best model fit was obtained with a trichromatic composite of wavelength sensitivities centered on UV (354 nm), blue (449 nm), and LW (527 nm) absorption maxima [68]. The same primary UV- and LW-sensitivity peaks were reported for *N. lugens,* as well as specific and pronounced attraction to blue light (470 nm) besides UV (365 nm, 385 nm) and green (525 nm) light [24]. Mirroring the situation in the Sternorrhyncha, these data are compatible with preserved trichromacy in *N. lugens* through compensatory neofunctionalization of the planthopper UV-der opsin paralog.

Another key requirement for color vision is the differential expression of opsin paralogs with different peak sensitivities in defined photoreceptor subtypes [34]. To date, retinal opsin gene expression data have been reported for *N. lugens* [24], *N. cincticeps* [68], and *A. pisum* [22]. Of these, *N. cincticeps* represents the ancestral state, possessing singleton homologs of the UV-, B-, and LW-opsin subfamilies [68]. In this species, the singleton LW-opsin homolog is expressed in a defined set of seven photoreceptors per ommatidium. The photoreceptor subtype not expressing LW-opsin is characterized by the seemingly stochastic but mutually exclusive expression of the UV- or B-opsin homolog [68]. Taken together, the retinal mosaic defined by the differential expression of LW-, B-, and UV-opsin is consistent with the behavioral and physiological evidence of trichromacy in *N. cincticeps* [68]. 

In the B-opsin lacking *N. lugens* and *A. pisum*, strong expression of the singleton LW-opsin was detected uniformly throughout the retina, suggesting expression in most, if not all, photoreceptors. The UV-opsin paralogs, in contrast, were detected in more selective patterns. In *N. lugens*, the candidate blue-shifted UV-der paralog (BPH_UVop1: XP_039285232) was detected in the ventral periphery of the compound eye while the ancestral UV-anc paralog (BPH_UVop2: XP_039285228) was detected in a subset of photoreceptors that were more evenly distributed across the retina [24]. In *A. pisum*, both the UV-der opsin (Ap-SWO2: XP_001951588) and the UV-anc opsin paralog (Ap-SWO3: XP_001951613) were detected in a similar low density punctuated distribution of photoreceptors across the retina [22], highly reminiscent of the retinal mosaic of *N. cincticeps* [66]. While the cellular resolution of the expression studies in *N. lugens* and *A. pisum* is too limited for conclusive inferences, the broader LW-opsin and the selective UV-opsin expression patterns in these species are consistent with the existence of LW-, B-, and UV-specific photoreceptors as in the ancestrally organized *N. cincticeps* [68].

### 4.4. Parallel vs. Convergent Events Leading to the Compensatory Replacements of B-Opsin with Peak Sensitivity Shifted UV-Opsins in Aphids and Planthoppers

Evolutionary change can lead to similar outcomes in different lineages by either equally congruent changes in homologous units or qualitatively unrelated changes in non-homologous units. The first type of scenario is commonly referred to as parallel evolution, while the second one represents convergent evolution. In the case of molecular units, like genes, similar outcomes can emerge in different lineages through the combination of parallel and convergent changes. Applied to the similar deployment of peak sensitivity modified UV-opsins for the replacement of B-opsins, the matching modification of homologous units constitutes parallel changes in aphids and planthopper evolution. At the protein sequence level, however, the number of changes in non-homologous sites is considerably higher than the two protein sequence sites that changed in parallel. Moreover, despite the very likely much younger age, the aphid UV-opsin sister paralogs diverged more dramatically based on the average pairwise evolutionary protein sequence divergence of 0.34 (±−0.013) than the planthopper UV-opsin sister paralogs with an average pairwise evolutionary protein sequence divergence of 0.19 (±−0.013) (Appendix A).

While these differences may be surprising given the similarity of the outcome, they are partly explained by the at least 300 million years of evolutionary separation between the aphid and planthopper lineages [8]. The second explanation must reside in the large space of protein sequence variation that can modify the endogenous UV peak sensitivity of the chromophore to blue. The parallel cases of B-opsin loss compensation by UV-opsin gene duplicates in aphids and planthoppers thus constitute a valuable resource for elucidating the protein sequence combinatorics underlying adaptive opsin evolution.

## Figures and Tables

**Figure 1 insects-14-00774-f001:**
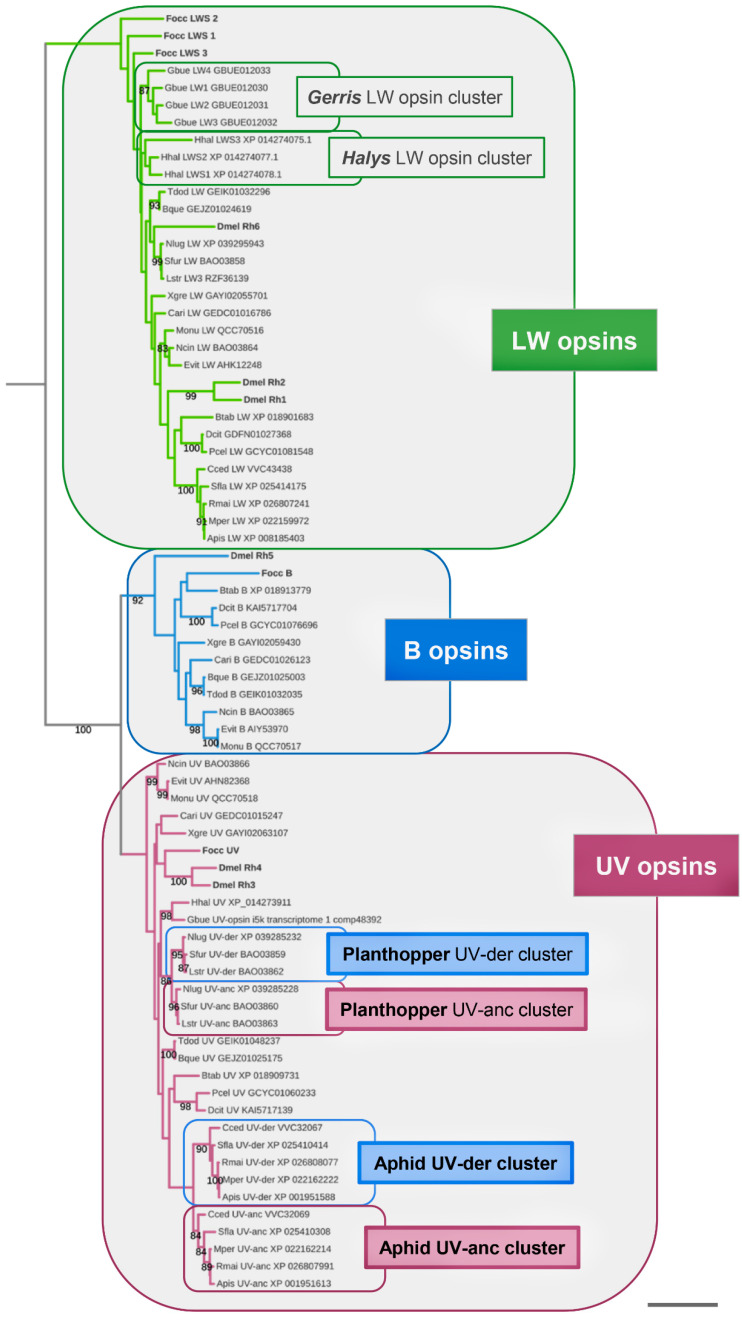
Global maximum likelihood gene tree of visual opsins from select hemipteran and outgroup species. Numbers at branches represent nonparametric bootstrap support from 100 replications. Scale bar corresponds to 0.25 substitutions per amino acid site. Opsin homologs from the outgroup species *Drosophila melanogaster* (Dmel) and *Frankliniella occidentalis* (Focc) indicated by dark grey bolt font. Abbreviations of included hemipteran species: Apis = *Acyrthosiphon pisum*, Btab = *Bemisia tabaci*, Bque = *Burbunga queenslandica*, Cari = *Clastoptera arizonana*, Dcit = *Diaphorina citri*, Cced = *Cinara cedri*, Evit = *Empoasca vitis*, Gbue = *Gerris buenoi*, Hal = *Halyomorpha halys*, Lstri = *Laodelphax striatellus,* Ncin = *Nephotettix cincticeps*, Monu = *Matsumurasca onukii*, Mper = *Myzus persicae*, Nlug = *Nilaparvata lugens*, Pcel = *Pachypsylla celtidismamma,* Rmai = *Rhopalosiphum maidis*, Sfla = *Sipha flava*, Sfur = *Sogatella furcifera*, Tdod = *Tamasa doddi*, Xgre = *Xenophysella greensladeae.*

**Figure 2 insects-14-00774-f002:**
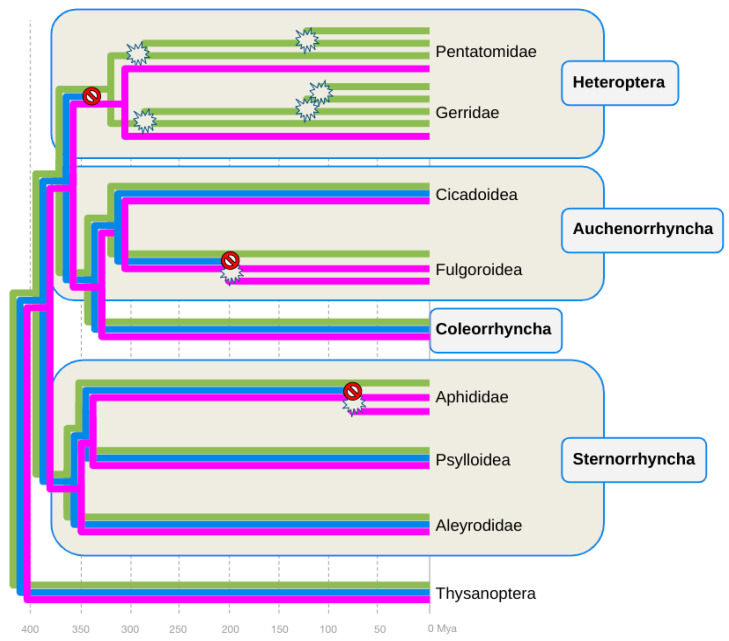
Gene species reconciliation tree of visual opsin gene family evolution in the Hemiptera. Phylogenetic relationships and clade ages based on [8]). Purple, blue, and green phylograms represent the UV-, B-, and LW-opsin subfamily gene trees. Stop signs indicate hypothesized time points of inferred B-opsin loss events. Burst signs indicate hypothesized time points of inferred UV-opsin duplication events. Mya = million years ago.

**Figure 3 insects-14-00774-f003:**
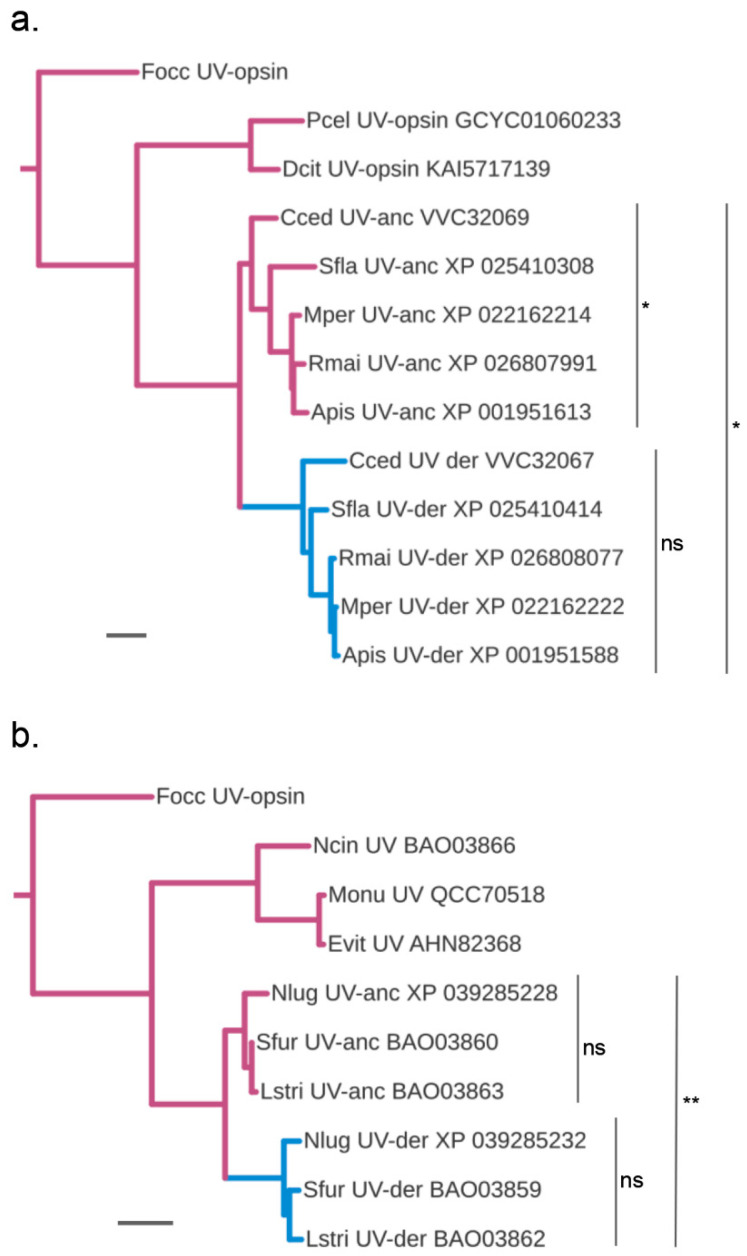
Protein sequence diversification rate differences between ancestral and putatively blue-shifted UV-opsin paralogs of planthoppers and aphids. Maximum likelihood gene trees of the (**a**) aphid and (**b**) planthopper UV-opsin clades. Purple and blue branches highlight the presumed ancestrally UV-sensitive, UV-anc, and blue-sensitivity-shifted UV-der paralog lineages, respectively. Scale bars represent 0.01 substitutions per site. ns = not significant, * significant at *p* < 0.05, ** significant and *p* < 0.005. See Figure 1 for species abbreviations. (**a**) Tested individually for molecular clock-compatible divergence rates, the UV-opsin homologs in the aphid UV-anc and UV-der paralog groups have diverged at significantly (*p* = 0.023) and non-significantly (*p* = 0.59) different rates. Clock-like protein sequence divergence is also rejected when the members of both clades are tested together (*p* = 0.026). The singleton UV-opsin homologs of *Pachypsylla celtidismamma* (Pcel), *Diaphorina citri* (Dcit), and *Frankliniella occidentalis* (Focc) (Thysanoptera) were included as outgroup sequences in the molecular clock tests. (**b**) Members of the planthopper UV-anc and UV-der paralog groups have not significantly diverged from other members within their paralog groups based on pairwise relative rate test results (*p* > 0.5). The singleton UV-opsin homolog of *Empoasca vitis* (Evit) was used as outgroup in the pairwise relative rate tests. However, clock-like protein sequence divergence is rejected when the members of both clades are tested together (*p* = 0.0068). The singleton UV-opsin homologs of *Nephotettix cincticeps* (Ncin), *Empoasca vitis*, *Matsumurasca onukii* (Monu), and *Frankliniella occidentalis* (Focc) were included as outgroup sequences in the molecular clock tests.

**Table 1 insects-14-00774-t001:** Genomic linkages of duplicated planthopper and aphid UV-opsin paralogs. Alternating white and grey backgrounds are used as visual dividers of species-specific rows.

Species	UV-Opsins	Locus ID	Linkage Group	Start (bp)	End (bp)	Interlocus Distance (bp)	Number of Intervening Genes
*Nilaparvata lugens*	XP_039285228	LOC111058697	5	75,381,302	75,410,087		
	XP_039285232	LOC111051439	5	75,416,027	75,437,887	5940	0
*Acyrthosiphon pisum*	XP_001951588	LOC100163348	A3	27,235,720	27,247,458		
	XP_001951613	LOC100161312	A3	27,217,337	27,226,019	30,121	0
*Myzus persicae*	XP_022162222	LOC111028006	NW_019101158	450,218	461,548		
	XP_022162214	LOC111027999	NW_019101158	432,490	442,835	29,058	1
*Rhopalosiphum maidis*	XP_026808077	LOC113550447	NC_040877	46,583,828	46,592,054		
	XP_026807991	LOC113550397	NC_040878	46,570,436	46,578,338	21,618	0

**Table 2 insects-14-00774-t002:** Candidate tuning site amino acid changes in aphid and planthopper UV-opsin paralogs. Site numbers represent homologous sites in bovine Rhodopsin (NP_001014890) or, when denoted with an asterisk, UV-opsin 2 (WCQ76394) of the jewel beetle species *Chrysochroa rajah* [54]. Sites that have been previously predicted or demonstrated to affect spectral sensitivity are indicated in bold print in column 1. Row cell triplets with light grey backgrounds indicate derived amino acid residue changes in putatively ancestrally UV-sensitive opsin paralogs (UV-anc). Row cell triplets with dark grey backgrounds and white bold font indicate derived amino acid residue changes in the putatively derived blue-sensitive UV-opsin paralogs (UV-der). Row cell triplets with bold black font indicate sites with amino acid residue changes in previously predicted or demonstrated tuning sites (90, 105, 293, 294). Ancestral amino acid residue states in aphids defined by amino acid residue states in the singleton UV-opsin orthologs of *P. celtidismamma* and *D. citri* (see Figure 3a). Ancestral states in planthopper UV-opsins defined by amino acid residue states in the singleton UV-opsin orthologs of *N. cincticeps*, *E. vitis*, *M. onukii* (see Figure 3b). See Appendix A for a complete table of investigated sites and Appendix A for annotated multiple sequence alignment. The uppercase letters correspond to the international amino acid single-letter code.

	Aphids			Planthoppers		
Site	Ancestral	UV-anc	UV-der	Ancestral	UV-anc	UV-der
15	** L **	** L **	** M **	L	L	L
29 *	P	A	P	** T **	** T **	** L **
33	Y	Y	Y	** Y **	** Y **	** H **
45 *	E	E	E	A	E	A
47 *	D	S	R	E	D	E
43	L	M	S	I/L	L	V
47	** F **	** F **	** L **	** F **	** F **	** Y **
52	V/I/L	L	V	** L **	** L **	** F **
63	L/I	C	C	** C **	** C **	** S **
84	F	F	F	** F **	** F **	** L **
85	** V/M **	** V/L/C **	** S **	M/L	M/L	L
87	** M **	** M **	** V **	M	M	M
**90**	**K**	**K**	**V**	K	K	K
94	** F **	** F **	** L **	F	F	F
102	** G **	** G **	** K(T) **	G	G	G
**105**	**Q**	**Q**	**P**	L	T/S	A
**125**	G/S	S	G	A	S	S
183	** Y **	** Y **	** F **	** Y **	** Y **	** F **
201	R	R,K	K	R	Q	R
204	V	L/V	V	** V **	** V **	** T **
244	(D)S	Q	Q	** Q **	** Q **	** M **
264	S	S/A	A	** S **	** S **	** A **
273	L	M	M(L)	L	M	L
286	** G **	** G **	** I **	G	G	G/V
**293**	C	I	V	**L**	**C**	**L**
**294**	C,T,A	F	F	**T**	**T**	**A**
295	** C **	** C **	** A **	C	C	C

## Data Availability

All data are available in publicly accessible repositories.

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
