# Peer review of "Parallel Losses of Blue Opsin Correlate with Compensatory Neofunctionalization of UV-Opsin Gene Duplicates in Aphids and Planthoppers"

_insects, 2023, doi:10.3390/insects14090774_

Round 1
Reviewer 1 Report
This study examines the losses and duplications of opsins across Hemiptera based on available genome data. In addition, putative spectral tuning sites of interest have been explored and discussed based on predictions of visual-pigment sensitivity (i.e. ancestral UV- or derived blue-sensitivity). The findings point to a complex scenario of losses and duplications across the group and potentially different opsin shifting mechanisms of UV duplicates, between suborders. I found this study to be very interesting and the manuscript well written. However, I think there should be improvement/greater clarification of the potential spectral tuning sites included.
Major comments
Please could the author mention how complete the genomes are that have been used for opsin searches. Is it possible that opsin copies have been missed in those taxa where no b-opsin has been recovered? If these are high quality genomes with good coverage then I think this is important to mention, as it lends more support to these findings.
I think a number of sites identified (e.g. 15, 47) may be located at a distance from the binding pocket and therefore may not interact with the chromophore. I think at least some of these distant positions are based on positive selection analyses from other studies and therefore could represent other non-spectral functional changes. I think it would be useful to either model the binding pocket or to point out those sites that are likely to be near the binding pocket, based on alignment with other species where the binding pocket has been predicted. Based on what is known currently, it may be unlikely that such distant sites affect spectral tuning. Similarly, I think it would be very useful to include the supporting evidence for each site in the table as it is not clear to me why they have been included, e.g site 47. If these sites are not likely to affect spectral tuning, rather alter non-spectral function, then the later results and discussion will need to be expanded or softened (e.g. L302).
General comments/suggestions
Please could Fig. 1 text be enlarged and the resolution increased as I found it difficult to read the tip labels.
In a few places in the text, the UV and blue sensitivity of duplicates is stated quite strongly. While I agree that this is compelling, based on the evidence presented, I would suggest softening some of the language (e.g. L15-16 and L29-30) as this has not yet been tested explicitly. Similarly, I think it would be beneficial to place some of the discussion where evidence for UV, blue and green sensitivity is described for aphids+ planthoppers in the introduction or include an introductory summary version. I think this would give better support earlier in the text for the designation of UV duplicates as putative UV and blue copies later on.
Minor comments
L70-73 Suggest splitting up or simplifying this sentence as it is quite complex.
L80 Is the author referring specifically to trichromatic colour vision? Dichromatic colour vision could be present in species with spectral channel losses.
L103 Suggest changing “an updated” to “a more extensive” or similar as I think this study covers more of the diversity than what has been previously explored.
Table 1 – I think column heading 2 should be “Suborder”.
Table 2 – Please could the author add some text to discuss the significance of the genomic linkage data that is presented here. I don’t think it is mentioned in the discussion.
L 216 Please could the author clarify substitution rate constancy in this context and how it relates to the interpretation of the figure 3 comparisons.
L498 Were outgroup (i.e. non-insect) opsins used to root the gene tree (Fig 1.)?
Author Response
Many thanks for your much-appreciated comments. Please see attached file for detailed responses.

Reviewer 2 Report
In this manuscript titled "Parallel losses of blue opsin correlate with compensatory neofunctionalization of UV-opsin gene duplicates in aphids and planthoppers," the author employed the bioinformatics analysis of visual opsin genes in Hemiptera. The study establishes B-opsin loss and correlates with UV-opsin duplications in the aphids and planthoppers. Except for the figures, which are pixelated, the manuscript is written clearly and suitable for publication.Using the existing insect genome database, the author reports that the B-opsin gene family lost at least three independent times during hemipteran diversification. Timing B-opsin losses correlated with duplication of UV-opsin. Though the reported study interests the broader audience, the analysis is limited to the sampling of insect families to support the conclusion. As the author suggested, the broader sampling with the availability of sequence data for additional insect taxa will strengthen the findings of this work. The author might consider the suggestions below to revise the manuscript:
1. Authors should provide a high-resolution image and clear labeling for easy readability. The figures are crowded and hard to read.
2. Discussing a parallel process – loss and duplication of gene families reported in insects and other species and the evolutionary implications would be interesting to a broader audience.
Author Response

(The authors gave the same response as above.)
